# RNA *In Situ* Hybridization for Detecting Gene Expression Patterns in the Abdomens and Wings of *Drosophila* Species

**DOI:** 10.3390/mps4010020

**Published:** 2021-03-10

**Authors:** Mujeeb Shittu, Tessa Steenwinkel, William Dion, Nathan Ostlund, Komal Raja, Thomas Werner

**Affiliations:** 1Department of Biological Sciences, Michigan Technological University, 740 Dow Building, Houghton, MI 49931, USA; mshittu@mtu.edu (M.S.); testeenw@mtu.edu (T.S.); nrostlun@mtu.edu (N.O.); 2Integrative Systems Biology Graduate Program, University of Pittsburgh School of Medicine, Biomedical Science, Tower 3, 3501 Fifth Avenue, Pittsburgh, PA 15213, USA; wad14@pitt.edu; 3Aging Institute of UPMC, University of Pittsburgh School of Medicine, Bridgeside Point 1, 5th Floor, 100 Technology Drive, Pittsburgh, PA 15219, USA; 4Department of Pathology and Immunology, Baylor College of Medicine, One Baylor Plaza, Houston, TX 77030, USA

**Keywords:** *Drosophila*, *in situ* hybridization, gene expression

## Abstract

RNA *in situ* hybridization (ISH) is used to visualize spatio-temporal gene expression patterns with broad applications in biology and biomedicine. Here we provide a protocol for mRNA ISH in developing pupal wings and abdomens for model and non-model *Drosophila* species. We describe best practices in pupal staging, tissue preparation, probe design and synthesis, imaging of gene expression patterns, and image-editing techniques. This protocol has been successfully used to investigate the roles of genes underlying the evolution of novel color patterns in non-model *Drosophila* species.

## 1. Introduction

RNA *in situ* hybridization (ISH) is a method used to detect and localize specific mRNA transcripts in cells, tissues, and whole organisms. Early ISH procedures used radioactively labeled RNA probes that hybridized with denatured DNA in tissues. The RNA-DNA hybrids were then detected by autoradiography [1,2]. A significant technical advancement to this method was the development of non-radioactive labeling systems that facilitated colorimetric visualization, which allowed for gene expression patterns to be observed in intact *Drosophila* embryos [3]. mRNA distribution patterns can now be detected by treating fixed tissues with digoxigenin-labeled RNA probes, to which F_ab_ fragments of anti-digoxigenin attached to alkaline phosphatase (anti-DIG-AP) are bound. The addition of 5-bromo-4-chloro-3-indoyl-phosphate (BCIP) and 4-nitroblue tetrazolium chloride (NBT) chromogenic solutions then results in the formation of a purple crystalline deposit, wherever the probe has bound to its target mRNA. The results, indicative of the gene expression patterns of interest, can be observed under a regular stereo microscope [3,4]. The non-radioactive ISH method is sensitive, easier, and safer.

ISH is a very powerful molecular tool used in research and diagnostics. The non-radioactive RNA ISH technique has been applied to studies in developmental biology, evolutionary biology, and cancer biology [5,6,7,8]. It has played a significant role in the detection of mRNA expression in humans, mice, insects, and in viral RNA detection [9,10,11,12,13,14].

Detailed protocols to perform RNA ISH in *Drosophila* embryos can be found in the published literature [4,15,16]. However, these protocols cannot be adapted for pupal wings and abdomens because additional critical steps are required for processing pupal tissues. Especially during the early pupal stage, RNA ISH can be very difficult to perform due to the fragility of the newly forming adult tissues. Although several studies have used the RNA ISH technique to detect and characterize gene expression patterns on pupal wings and abdomens of various *Drosophila* species [6,7,17,18,19], no protocol describing this technique in *Drosophila* pupae has been published in any scholarly journal.

Here we provide a protocol suitable for model and non-model *Drosophila* species alike, detailing the steps of probe design and synthesis, pupal tissue dissection, the ISH process, and imaging techniques. We have applied this protocol to investigate the gene-regulatory networks governing the development and evolution of pigmentation patterns in pupal abdomens and wings of a variety of *Drosophila* species, such as *D. melanogaster*, *D. guttifera*, *D. deflecta*, *D. palustris*, and *D. subpalustris* [5,6,20]. The users of this protocol should pay close attention to the pupal tissue processing steps, as the outcome of an RNA ISH critically depends on the tissue quality.

### 1.1. Overview

The major steps involved in ISH for *Drosophila* abdomens and wings are outlined in the flowchart (Figure 1). First, species-specific PCR primers are designed to amplify a partial coding region of the gene of interest, using genomic DNA as a template. The PCR product is then cloned into the vector pGEM^®^-T Easy. A region containing the PCR product with flanking T7/Sp6 sites is PCR-amplified from this vector and used as a template for the synthesis of the DIG-labeled antisense RNA probe by *in vitro* transcription. Abdominal epidermis preparations involve pupal dissection, the removal of unwanted tissues by pipetting, fixation of the epidermal cell layer, and tissue storage. For wing preparations, the order of wing dissection and fixation depends on the pupal stage. In younger pupae, the wings are fixed before they are separated from the body, while in older pupae, the fixation step follows the separation of the wings. The important steps in the ISH procedure provided here are xylene treatment, tissue rehydration, fixation, proteinase K treatment, second fixation, prehybridization, probe addition, anti-DIG-AP F_ab_ treatment, and NBT/BCIP staining.

### 1.2. Application of This Protocol

ISH has been integral to the field of evolutionary developmental biology in that it is a valuable technique to study the emergence of novel traits, such as the color patterns of butterflies and fruit flies [17,18,21,22]. This protocol allows for the detection of gene expression patterns in abdomens and wings of different *Drosophila* species. We have used this procedure in model and non-model *Drosophila* species to study toolkit and terminal pigmentation genes involved in color pattern formation, such as *D. melanogaster*, *D. guttifera*, *D. deflecta*, *D. palustris*, and *D. subpalustris* [5,6,20]. No specialized skills are required to use this protocol; we have had undergraduate students generate high-quality results in our lab. We believe that this protocol will facilitate the study of novel gene expression patterns in rare and unstudied fruit flies, which can be collected and brought into the lab using the two new field guides to drosophilid species [23,24].

### 1.3. Advantages and Limitations

Our protocol allows for the detection of developmental toolkit genes’ involvement in color patterning in *Drosophila* species. The most important advantage of this ISH protocol is that it enables the detection of gene expression patterns in very early pupal abdomens (as soon as the epidermal layer has formed, i.e., from pupal stage P7 onwards), as well as early wings that cannot be dissected without prior fixation (from stage P5ii onwards) (Figure 2). This procedure requires basic laboratory equipment to generate high-quality ISH results [5,20]. However, this method is not without limitations. One of the limitations of the ISH technique is that it is semi-quantitative, and although it can detect and visualize spatial gene expression patterns, it is less accurate in determining quantitative gene expression differences than RNA-seq-based methods. Also, the final staining outcome of an ISH technique depends on the duration of the staining reaction, probe concentration, and fixation conditions, therefore making it sometimes difficult to generate the desired result in one attempt. Furthermore, the development of background staining is a limiting factor for how long a staining reaction can continue, which causes problems when a gene is weakly expressed.

## 2. Experimental Design

### 2.1. Probe Design and Synthesis

The process starts with designing PCR primers to amplify a partial protein-coding region from a single exon, using the GenePalette software [26] as described in Appendix B. Primers are 18–25 bases in length and are designed to yield products of 200–500 bp in size. We prepared and used external primers to amplify the PCR product and internal primers to amplify a shorter “internal PCR product”. When comparing expression patterns of a gene among different *Drosophila* species, we used the multiple alignment tool in GenePalette to identify the most highly conserved regions of the open reading frame. For the probe design, we chose a region that did not contain indels among the species; thus, the PCR products for the same gene for different species would have the same length. It should be noted that the use of the same PCR primers that are designed from a highly conserved exon to amplify the amplicon in different *Drosophila* species may lead to mismatches in the sequence, which can reduce the hybridization efficiency of the probe. The PCR products (inserts) were then cloned into the vector pGEM^®^-T Easy, which contains the Sp6 and T7 promoters required for *in vitro* synthesis of antisense RNA probes. The *E. coli DH5-α* competent cells were transformed with the cloned vector in order to generate several clones, after which we performed the colony PCR to isolate the positive colonies. The positive colonies were cultured overnight, and the cloned plasmids were extracted (mini-prep DNA). In order to determine the orientation of the insert in the vector, we carried out insertion direction PCR. Depending on the direction of insertion, we used the Sp6/T7 RNA polymerase to synthesize antisense RNA probes.

### 2.2. Pupal Staging

We collected wandering third-instar (L3) larvae in a 10 cm Petri dish with moist tissue paper covering the bottom. After the larvae began to pupate, we followed the progression of the pupal stages under a dissection scope. We adopted the description by Bainbridge and Bownes [25] and Fukutomi et al. [27] for *Drosophila* pupal stage determination (Figure 2).

### 2.3. Abdominal Epidermis Preparation

Once the pupae reached the desired stage, they were placed in groups of ten onto a double-sided tape fixed to a microscope slide to cut them and clean the epidermal tissue. We performed two types of cuts: (1) the pupae were placed with their ventral side facing the tape and then cut longitudinally between both eyes (dorsal cut) and (2) one of their lateral sides faced the tape, and the cut ran longitudinally through the pupae, separating the dorsal from the ventral half (lateral cut) (Appendix A). Dorsal cuts are best suited to examine lateral patterns of gene expression, while lateral cuts allow for the visualization of the dorsal and ventral regions of the abdomen. The cut pupae were then washed with phosphate-buffered saline (PBS), fixed in paraformaldehyde, and stored in 100% ethanol at −20 °C until further processing by ISH.

### 2.4. Pupal Wing Preparation

Wing preparations are possible from stage P5ii onward. It is important to note that wings from stages P5ii to P8 are too fragile to be dissected without prior fixation. These early pupae were pulled out of their puparia in PBS with a pair of fine forceps, followed by cutting off the head and the tip of the abdomen with a small pair of surgical scissors. The resulting carcass tubes were cleaned by pipetting PBS through them to remove the inner organs. The carcasses were then fixed in paraformaldehyde (see Section 6 for reagent preparation), overnight at 4 °C, after which the hardened wings were dissected from the pupae with two pairs of fine forceps. Pupae of stage P9 and older have sturdier wings, which allows the fixation process to follow dissection, instead of preceding it. These older pupae were placed with the ventral side down on a glass slide with double-sided tape. The pupae were pulled out of their puparia by the head and submerged in a glass-viewing dish in distilled water. The wings were then carefully extracted from the pupal membrane and allowed to inflate, followed by fixing them with paraformaldehyde at room temperature for 30 min. Regardless of the pupal stage, the extracted wings were washed twice with methanol and twice with 100% ethanol after the fixation step and then stored in ethanol at −20 °C until the ISH procedure was performed.

### 2.5. ISH of Drosophila Abdomens and Wings

On the first day of the ISH, the processed tissues (pupal abdomens or wings) were carefully transferred into the wells of a glass-viewing dish, using a cut 1 mL pipette tip. They were then treated with xylenes to remove any fatty tissue, re-hydrated, fixed, washed, treated with proteinase K, washed, post-fixed, and prehybridized. After the prehybridization, DIG-labeled RNA probes were added, and the samples were incubated at 65 °C for 18 h to 3 days (d). On the second day, any unhybridized probe molecules were washed away to reduce background staining. The washed tissues were incubated in anti-DIG-AP F_ab_ fragments at 4 °C overnight. On the third and final day, the tissues were treated with NBT/BCIP staining solution to detect the dark-purple hybridization pattern. This reaction took place in the dark. The staining progress was observed under a dissecting scope every 20 min to avoid overstaining. After the staining reached its desired intensity, the staining solution was washed off with staining buffer, then with PBT, and the gene expression patterns were observed under a stereo microscope. Several images of the tissues were taken at slightly different focal planes and Z-stacked with Helicon Focus software. A minimum of three days is required to perform the ISH experiment in *Drosophila* using this protocol.

### 2.6. Materials

Distilled H_2_OTaq 2× MeanGreen Master Mix (Syzygy Biotech, Grand rapids, MI, USA, www.integratedscientificsolutions.com, (accessed on 9 March 2021))Agarose (Dot Scientific Inc., Burton, MI, USA, cat. no. AGLE500)TAE buffer (see Section 6)1 mM dATP (Sigma-Aldrich, St. Louis, MO, USA, https://www.sigmaaldrich.com/ (accessed on 8 March 2021))10× PCR buffer (Sigma-Aldrich, St. Louis, MO, USA, cat. no. P2192)Taq Polymerase (Thermo Scientific, Waltham, MA, USA, cat. no. EP0401)Gel extraction kit (Thermo Scientific, Waltham, MA, USA, cat. no. K0692)pGEM^®^-TEasy vector system (Promega, Madison, WI, USA, cat. no. A1360, store at −20 °C)*DH5-α* competent cells (Thermo Scientific, Waltham, MA, USA, cat. no. EC0112)LB medium (see Section 6)Culture media preparation (see Section 6)Ampicillin (Sigma-Aldrich, St. Louis, MO, USA, cat. no. A0166, store at −20 °C)IPTG (Fisher Scientific, Waltham, MA, USA, cat. no. BP1755-1, store at −20 °C)X-GAL (Sigma-Aldrich, St. Louis, MO, USA, cat. no. 7240-90-6, store at −20 °C)Plasmid mini-prep kit (Thermo Scientific, Waltham, MA, USA, cat. no. K0503)M13F and M13R vector primers (Integrated DNA Technologies, www.idtdna.com, (accessed on 9 March 2021)) (see Appendix A for primer sequences)DIG RNA labeling kit (SP6/T7) (Roche, Basel, Switzerland, cat. no. 11175025910, store at −20 °C)DNA gel-loading buffer (6X) (Thermo Fisher Scientific, Waltham, MA, USA, cat. no. R0611 store at 4 °C)Linear acrylamide (AMRESCO, Cleveland, OH, USA, cat. no. K548, store at 4 °C)Sodium acetate (Sigma-Aldrich, St. Louis, MO, USA, cat. no. 127-09-3)100% Ethanol (DECON Laboratories Inc, King of Prussia, PA, USA, cat. no. 64-17-5)Hybridization buffer (see Section 6)PBS (see Section 6)Fixation buffer (see Section 6)Xylenes (Sigma-Aldrich, St. Louis, MO, USA, cat. no. 534056) CAUTION Harmful by inhalation and in contact with skin.Methanol (Sigma-Aldrich, St. Louis, MO, USA, cat. no. 38460) CAUTION Methanol is poisonous. It should not be inhaled, swallowed or be allowed to touch the skin.PBT (see Section 6)Proteinase K (Sigma-Aldrich, St. Louis, MO, USA, cat. no. P2308, store at −20 °C)Anti-digoxigenin-AP Fab fragment (Roche, Basel, Switzerland, cat. no. 11093274910, store at 4 °C)Staining buffer (see Section 6)Staining solution (see Section 6)NBT (Promega, Madison, WI, USA, cat. no. S380C, store at store at −20 °C) CAUTION Toxic.BCIP (Promega, Madison, WI, USA, cat. no. S381C, store at 4 °C) CAUTION Toxic.Trizma Base (Sigma-Aldrich, St. Louis, MO, USA, cat. no. T6066)Glacial acetic acid (Sigma-Aldrich, St. Louis, MO, USA, cat. no. 320099)EDTA (Sigma-Aldrich, St. Louis, MO, USA, cat. no. E0399)37% HCl (Sigma-Aldrich, St. Louis, MO, USA, cat. no. 320099) CAUTION Toxic when inhaled, causes irritation to the respiratory tract, and causes skin burn.Tryptone (Sigma-Aldrich, St. Louis, MO, USA, cat. no. T7293)Yeast extract (Sigma-Aldrich, St. Louis, MO, USA, cat. no. Y1625)NaCl (Sigma-Aldrich, St. Louis, MO, USA, cat. no. S3014)NaOH (Sigma-Aldrich, St. Louis, MO, USA, cat. no. S5881) CAUTION Causes severe skin burns and eye damage.Agar (Sigma-Aldrich, St. Louis, MO, USA, cat. no. A6686)Formamide (Sigma-Aldrich, St. Louis, MO, USA, cat. no. 221198) CAUTION Suspected of causing cancer.Salmon sperm DNA (Invitrogen, Carlsbad, CA, USA, cat. no. 15632-011)Heparin sodium salt (Sigma-Aldrich, St. Louis, MO, USA, cat. no. H3393)Tween 20 (Sigma-Aldrich, St. Louis, MO, USA, cat. no. P1379)Triton X-100 (Sigma-Aldrich, St. Louis, MO, USA, cat. no. X100)K_2_HPO_4_ (Sigma-Aldrich, St. Louis, MO, USA, cat. no. P2222)KH_2_PO_4_ (Sigma-Aldrich, St. Louis, MO, USA, cat. no. P5655)Sodium citrate (Sigma-Aldrich, St. Louis, MO, USA, cat. no. S1804)Deoxycholic acid (Sigma-Aldrich, St. Louis, MO, USA, cat. no. D2510)16% Paraformaldehyde (Electron Microscope Sciences, Hatfield, PA, USA, cat. no. 15710) CAUTION Causes skin and eye irritation. Suspected of causing genetic defects and may cause cancer.MgCl_2_ (Sigma-Aldrich, St. Louis, MO, USA, cat. no. M8266)Gel slick (Lonza, Basel, Switzerland, cat. no. 50640).

### 2.7. Equipment and Supplies

Thermocycler (Eppendorf AG, Hamburg, Germany, cat. no. 6325)Incubator (Thermo Fisher Scientific, Waltham, MA, USA, cat. no. 6246)Centrifuge (Eppendorf AG, Hamburg, Germany, cat. no. 5424)Refrigerated centrifuge (Eppendorf AG, Hamburg, Germany, cat. no. 5404R)Water bath (Thermo Scientific, Waltham, MA, USA, cat. no. 1521038, model HAAKE S3)Incubator shaker (New Brunswick, Edison, NJ, USA, model I-SERIES 24)Vortex mixer (Labnet, www.labnetinternational.com, (accessed on 9 March 2021), cat. no. S0100, model VX100)Electrophoresis power supply (Fisher Scientific, Waltham, MA, USA, cat. no. FB1000)Dissecting scope (Olympus, www.olympus-lifescience.com, (accessed on 9 March 2021), cat. no. SZ51)Digital heating block (Apollo Instrumentation, city, state abbrev if USA, country)Stereo Microscope (Olympus, www.olympus-lifescience.com, (accessed on 9 March 2021), model SZX16)DP72 camera (Olympus, www.olympus-lifescience.com, (accessed on 9 March 2021), model U-TV1X-2)UVP Transilluminator (BioDoc-It^TM^ Imaging System, www.uvp.com, (accessed on 9 March 2021))Digital monochrome printer (Mitsubishi, www.uvp.com, (accessed on 9 March 2021), model P95DW)Pipettors (Fisherbrand, Waltham, MA, USA, cat. no. 14-388-100)Blunt forceps (Dumont #2) (Fisher Scientific, Waltham, MA, USA, cat. no. 50-822-406)Fine forceps (Dumont #5) (Fisher Scientific, Waltham, MA, USA, cat. no. 50-822-450)Surgical scissors (Fine Science Tools, www.finescience.com/en-US/, (accessed on 9 March 2021) cat. no. 91500-09)1.5 mL Eppendorf tubes (Fisherbrand, Waltham, MA, USA, cat. no. 05-408-141)2 mL Eppendorf tubes (Fisherbrand, Waltham, MA, USA, cat. no. 05-408-132)Falcon tubes (Fisher Scientific, Waltham, MA, USA, cat. no. 14-959-70C)20 mL Scintillation vial (DWK Life Sciences, Wheaton, Millville, NJ, USA, cat. no. 74510-20)20 mL glass centrifugation tube (Pyrex, city, Corning, NY, USA, cat. no. 9825)Glass-viewing dish (Pyrex spot plates 9 concave depressions 22 mm O.D. × 7 mm deep) (Fisher scientific, Waltham, MA, USA, cat. no. 13-748B)Medium-sized Petri dish (VWR, Cleveland, OH, USA, cat. no. 25384-302)Microscope slides 75 × 25 × 1 mm (VWR Vistavision, Cleveland, OH, USA, cat. no 16004-368)Permanent double-sided tape (Scotch, www.scotchbrand.com, (accessed on 9 March 2021) cat. no 38-8507-5367-3)Clean razor blade (VWR, Cleveland, OH, USA, cat. No. 55411-050)Medium-sized paintbrush (Liner 44,278 Plaid^®^Xuancheng, Xuancheng, China)Kleenex paper (Fisher Scientific, Waltham, MA, USA, cat. no. 06-666-11)Kimwipes^®^ disposable wipers (Sigma Aldrich, St. Louis, MO, USA, cat. no. Z188964)Helicon Focus software (http://www.heliconsoft.com/heliconsoft-products/helicon-focus/ (accessed on 8 March 2021)).

## 3. Procedure

### 3.1. A-Tail Genomic PCR

Timing 4 h

(1)Prepare a reaction mix, according to the Table 1 below.(2)Amplify the PCR product according to the appropriate cycling conditions (Table 2).(3)Run the PCR product through a 1% (*w/v*) agarose gel in 1× TAE buffer by electrophoresis and visualize it under UV light.(4)If the size of the PCR product matches the expected size, perform a gel extraction.

### 3.2. Gel Extraction and Purification of PCR Products

Timing 30 min

(5)On a table-top UV light, cut out the gel slice containing the DNA fragment, using a clean razor blade and place it into a 2 mL Eppendorf tube.(6)Add 1:1 volume of binding buffer to the gel slice (*w/v*).(7)Incubate at 60 °C until the gel slice is completely dissolved.(8)Transfer the solubilized gel solution to the purification column. Centrifuge for 1 min and discard the flow-through.(9)Add 700 µL of wash buffer to the column. Centrifuge for 1 min and discard the flow-through.(10)Centrifuge the empty column for 1 min to completely remove the wash buffer.(11)Transfer the column into a 1.5 mL Eppendorf tube. Add 30 µL of elution buffer and incubate at room temperature for 1 min.(12)Centrifuge for 1 min to collect the DNA fragment in the 1.5 mL tube.(13)Store the purified DNA at −20 °C.


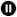
**PAUSE STEP** Purified PCR products can be stored at −20 °C for one month.


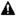
**CRITICAL STEP** MeanGreen master mix adds A-tails to the PCR products. In case you use a PCR master mix that does not add A-tails, add the A-tails after per forming the gel extraction, according to the Table 3 below. Also, note that A-tails de-grade after one month of storage at −20 °C.

(14)Incubate in a thermocycler at 72 °C for 45 min. Store at −20 °C.

### 3.3. Ligation

Timing 18 h (overnight)

(15)To ligate the A-tailed PCR product with the pGEM^®^-T Easy vector, use the reaction mix in the Table 4 below.

(16)Incubate the ligation reaction at 4 °C overnight.


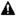
**CRITICAL STEP** The pGEM^®^-T Easy vector features T-overhangs essential for T/A cloning of an A-tailed DNA fragment. Also, this vector contains the SP6/T7 promoters, which will later flank the insert after an additional PCR reaction.

### 3.4. Transformation of DH5-α Cells and Colony PCR

Timing 18 h (transformation) and 4 h (colony PCR)

(17)Add 2 µL of the raw ligation product to 50 µL of *E. coli DH5-α* competent cells in a 1.5 mL Eppendorf tube.(18)Place the tube on ice for 45 min.(19)Heat shock the mixture at 42 °C for 1 min in a water bath.(20)Immediately transfer the tube back to the ice and leave it for 5 min.(21)Add 200 µL of LB medium to the cells.(22)Incubate the culture at 37 °C in an incubator shaker for 1 h at 200 r.p.m.(23)Prepare an ampicillin agar (200 µg/mL) in a medium-sized culture medium plate (see Section 6). The plate should be pre-made at least a day before use and stored at 4 °C.(24)Mix 100 µL of the IPTG and 50 µL of the X-Gal solutions and spread them evenly on the bacterial agar plate for the blue/white colony selection.(25)Add 50–100 µL of the bacterial culture (step 22) onto the agar plate.(26)Incubate the plate at 37 °C for 18 h.(27)Pick 12 white colonies with a small pipette tip (use fresh pipette tips to pick each colony) and suspend each bacterial colony in 10 µL of dH_2_O in 1.5 mL Eppendorf tubes. The white colonies should contain inserts, while the blue colonies likely contain empty, self-ligated vector.(28)Immediately perform a colony PCR to confirm the presence of the correct insert in these clones, as tabulated below (Table 5). Use the cycling conditions described in (Table 6).

### 3.5. Culturing the Positive Colonies

Timing 19 h

(29)Mix 6 µL of ampicillin (2 mg/mL) with 3 mL LB medium in a sterile glass tube.(30)Inoculate the tube with 5 µL of the bacterial colony suspension (Step 27).(31)Mix thoroughly.(32)Incubate at 37 °C in an incubator shaker for 18 h at 200 r.p.m.

### 3.6. Plasmid Extraction from a Positive Clone (Mini-Prep) Using the Plasmid Mini-Prep Kit by Thermo Scientific

Timing 40 min

(33)Pour ~1.8 mL of the cultured cells (containing the cloned plasmid) into a 2-mL Eppendorf tube.(34)Centrifuge at 15,000× *g* for 30 s at room temperature and discard the supernatant.(35)Add 250 µL of the resuspension buffer and vortex to resuspend the cell pellet.(36)Add 250 µL of the lysis buffer and mix by inverting the tube 10 times.(37)Add 350 µL of the neutralization buffer and mix by inverting the tube 10 times.(38)Centrifuge for 5 min at 15,000× *g* at room temperature.(39)Pour the supernatant into the spin column and centrifuge for 1 min at 15,000× *g* at room temperature.(40)Add 500 µL of the wash buffer to the spin column. Centrifuge for 1 min at 15,000× *g* at room temperature and discard the flow-through. Repeat this step.(41)Transfer the spin column into a clean 1.5-mL Eppendorf tube.(42)Add 30 µL of the Elution buffer to elute the plasmid DNA. Incubate for 2 min at room temperature and centrifuge for 2 min at 15,000× *g* at room temperature.(43)Store the mini-prep DNA (containing the cloned-plasmid) at −20 °C.


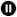
**PAUSE STEP** The DNA can be stored at −20 °C indefinitely.

### 3.7. Insertion Direction PCR

Timing 3 h

(44)Carry out the insertion direction PCR for each mini-prep DNA to determine the orientation of the insert in the pGEM^®^-TEasy vector to choose the correct RNA polymerase that will synthesize an antisense probe, according to Table 7.(45)Set up two PCR reactions simultaneously for each DNA clone, using the following primer pairs: (i) the M13F primer plus the gene-specific internal forward primer; and (ii) the M13F primer plus the gene-specific internal reverse primer.

(46)Perform the PCR reactions according to the cycling conditions described in (Table 8).(47)Perform gel electrophoresis of the PCR products, using a 1% (*w/v*) agarose gel in 1× TAE and visualize the bands under UV light.


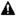
**CRITICAL STEP** If the primer pair M13F/internal reverse shows a PCR band, use Sp6 polymerase to make an anti-sense probe. However, if the primer pair M13F/internal forward shows a PCR band, use T7 polymerase to make an anti-sense probe. Only one of the primer pairs should produce a clear PCR band.

### 3.8. RNA Probe Synthesis

Timing 5 h

(48)PCR-amplify the cloned insert, using the mini-prep DNA as a template (Step 43). Use the M13F and M13R primer pair, as tabulated below (Table 9).

(49)Amplify according to the cycling conditions described in Table 8.(50)Run the electrophoresis of the PCR product, using a 1% (*w/v*) agarose gel in 1× TAE.(51)Extract the DNA band and elute it in 30 µL Elution buffer, as described in Steps 5–13.(52)Measure the DNA concentration.


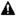
**CRITICAL STEP** For a high probe yield, use 0.05–0.1 µg/µL of the DNA as a template for the *in vitro* transcription reaction.

(53)Prepare the anti-sense RNA probe reaction mix as tabulated below (Table 10).

(54)Incubate at 37 °C for 2 h.


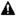
**CRITICAL STEP** Remember to use the correct RNA polymerase based on the insertion direction PCR result (Steps 44–47). (Recall: If the primer pair M13F/internal reverse shows a PCR band, use Sp6 polymerase to make an anti-sense probe. However, if the primer pair M13F/internal forward shows a PCR band, use T7 polymerase to make an anti-sense probe).

### 3.9. RNA Probe Quality Check and Precipitation

Timing 2 h

(55)Check the quality of the synthesized RNA probe by running 1 µL of the newly synthesized probe (Step 54) alongside 1 µL of the purified PCR (Step 51) on a gel.(56)Add 2 µL of the gel loading buffer into 9 µL of H_2_0, then add 1 µL of the probe.(57)Vortex, then briefly spin down.(58)Load the 12 µL on a 1% agarose gel and perform electrophoresis in 1× TAE buffer. An example of a gel image showing a successfully transcribed probe is shown in (Figure 3).(59)Precipitate the remaining 9 µL of the synthesized probe as shown below (Table 11).

(60)Incubate at −20 °C for 20 min.(61)Spin at 15,000 r.p.m at 4 °C for 30 min.(62)Pipette off the supernatant and discard.(63)Air-dry the probe-containing pellet for 5 min.(64)Dissolve the pellet in 50–100 µL of the pre-hybridization buffer on ice by carefully pipetting the liquid up and down.


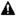
**CRITICAL STEP** Depending on the probe yield, use 100 µL of pre-hybridization solution to dissolve the pellet when you see a solid probe band on the gel or use 50 µL if the band is rather faint.


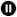
**PAUSE STEP** The RNA probe can be stored aliquoted for about two years at −20 °C (create a stock and several aliquots).

### 3.10. Drosophila Pupa Collection and Processing for Abdominal ISH

Timing 4 h


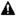
**CRITICAL STEP** The following instructions are critical for Steps 65–154: (1) Look through the dissecting scope during all pipetting steps. (2) Washes are with 1 mL of organic solutions for 5 min, and 10 min for aqueous solutions, unless otherwise indicated. (3) Washes involving hybridization solution are 500 µL. (4) Do not agitate the samples too much during washes; only gently move the liquid in and out of the pipette when removing liquid. (5) After washes, remove waste from the wells of the glass-viewing dish in 150 µL increments and with the same 200 µL tip. (6) Successive washes are done under a running clock.

(65)Collect wandering third-instar (L3) larvae from the *Drosophila* culture bottle and place them in a Petri dish with moist Kleenex paper on the bottom.(66)Store the Petri dish in a species-specific moist chamber so that different species are not mixed, and the moist chamber prevents the larvae from drying out.(67)Wait until the pupae are at the desired stage (Figure 2).(68)Use a dissecting scope set to 20× magnification to clearly see the key features of the pupal stages.(69)Prepare a glass slide with a piece of double-sided tape on it (dissection platform).(70)Fill the well of a glass-viewing dish with 1 mL of freshly prepared 1× PBS.(71)Use blunt forceps (type #2) to gently remove the pupae from the Petri dish (one at a time) and immediately transfer them onto the dissection platform.(72)Lay the pupae with their ventral side facing the tape and cut longitudinally between both eyes (dorsal cut) or lay them on their lateral side and cut longitudinally through the pupae, separating the dorsal from the ventral half (lateral cut) (Appendix A). Perform only one type of cut in a session.(73)With a razor blade, immediately cut each pupa lengthwise (starting with the one first placed on the tape). This is best accomplished with a single rapid cut from the anterior to the posterior end of the pupae. Ensure that the puparium is intact after cutting.(74)Using a medium-sized paintbrush, transfer a small amount of 1× PBS from the glass-viewing dish to each cut pupa to dissolve them from the tape.(75)Transfer the pupal halves with the brush into the well of a glass-viewing dish filled with 1× PBS.


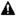
**CRITICAL STEP** Dissect 10 pupae within 2 min and then transfer them into 1× PBS immediately. Label each end of the razor blade and use each end to dissect 50–60 pupae, after which the blade is too blunt and should be discarded.

(76)With a pair of surgical (sharp-pointed) forceps (type #5), grasp an individual pupa half anteriorly (by the head) and gently wash away the internal organs with 1× PBS.(77)Use a pipettor to gently flush 1× PBS over the internal organs without touching the epithelial layer of the pupa with the pipette tip (Appendix A). Prevent the epidermal tissue from becoming detached from the puparium at this time, as the puparium provides mechanical protection throughout the process.


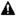
**CRITICAL STEP** You must ensure to keep the epithelial cell layer intact throughout the washing steps. Therefore, apply low pressure from the pipettor by setting a 20-µL pipettor to 8.5 µL for pupal stages P7, P8 or to 15 µL for pupal stage P9. For stages P10 and older, use a 200-µL pipettor set to 25 µL. Too much pressure or excessive washing will lead to the loss of epithelial cells.

(78)Immediately transfer the washed pupa halves into a well with 1× PBS.(79)Remove the 1× PBS solution from the well.(80)Add 1 mL of the fixation buffer for abdominal ISH (Section 6) to the pupa halves.(81)Incubate at room temperature for 1 h.(82)Rinse the fixed pupae 3 times with 1× PBS.(83)The fixed samples may be used to perform ISH immediately (jump to Step 109) or stored for later use. If storage is the goal, dehydrate the pupa halves through a dilution series of 1× PBS:100% ethanol (3:1, 1:1, 1:3) for 20 min in each solution at room temperature.(84)Rinse once and wash once in 1 mL 100% ethanol. With a cut 1 mL tip, transfer the pupae into a 2-mL Eppendorf tube and store in 100% ethanol at −20 °C.


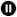
**PAUSE STEP** The processed pupae can be stored at −20 °C for 1 year.

### 3.11. Drosophila Pupa Processing for Wing ISH

Processing pupae of stages P5ii–P8

(85)Collect wandering larvae in a Petri dish with a wet tissue paper on the bottom.(86)Check the time when most puparia have formed and collect P5ii–P8 pupae.(87)Place a 15-mL and a 50-mL Falcon tube with 1× PBS on ice.(88)Ensure the glass-viewing dish is coated with gel slick.(89)In a glass-viewing dish, put a batch of 5 pupae into 1× PBS and take them by the head out of their puparium. The heads may get destroyed, which is fine.(90)Cut off the heads and the tip of the abdomens without squeezing the body (avoid liquid getting pushed into the wings). Then, hold the pupae by the thorax and use a 200 µL pipette to carefully suck and blow out the guts through the open abdomen.(91)Collect the empty carcasses in a 2-mL Eppendorf tube filled with 1.5 mL of fixation buffer on ice.(92)Clean out the glass-viewing dish with distilled water and proceed with the next batch of 5 pupae.(93)Fix overnight at 4 °C. On the next morning, place the tube with the fixed carcasses on ice.(94)Pipette about 5 carcasses into a glass-viewing dish and remove the pupal membrane from the wings.(95)Carefully rip off the wings with a small piece of thorax still attached and collect them in a scintillation vial containing about 4 mL of methanol at room temperature.(96)After all wings are dissected, pipette them all back into the clean glass-viewing dish and wait additional 5 min.(97)Wash 2× with methanol.(98)Wash 2× with 100% ethanol.(99)Store the wings in 100% ethanol at −20 °C.

Processing pupae of stage P9 and older

(100)Cut pupal wings in dH_2_O at room temperature in a glass-viewing dish coated with gel slick.(101)Allow the wings to inflate.(102)Place the wings in 1.5 mL of fixation on ice in a 2-mL Eppendorf tube for 30 min.(103)Wash 2× with methanol.(104)Wash 2× with 100% ethanol.(105)Store the wings at −20 °C in 100% ethanol.

**CAUTION** Allow methanol waste to evaporate in the hood or pour it down the drain and flush for 1 min. Do not inhale, swallow, or allow methanol to touch your skin.

### 3.12. ISH of Drosophila Abdomens and Wings

Timing 3 d


**Day 1**


(106)Take the processed wings and the pupa halves from the −20 °C freezer.(107)Transfer the pupa halves with a pipette and a cut 1-mL tip into a glass-viewing dish.(108)Transfer the wings with a pipette and a cut 200-µL tip into a glass-viewing dish. Ensure that the glass-viewing dish is gel slick coated (Appendix D).(109)Wash once with 100% ethanol.(110)Incubate for 30 min with 1mL 1:1 xylenes:ethanol (*v/v*) in a fume hood.(111)Rinse once and wash 5× with 100% ethanol.(112)Wash 2× with methanol (wings only).

**CAUTION** Incubate and discard all xylenes-containing washes in the fume hood. Xylenes can constitute a serious health hazard.

(113)Rehydrate the tissues through a dilution series of 1× PBS:100% ethanol (1:3, 1:1, 3:1) (*v/v*) and incubate at room temperature for 20 min in each solution (skip this step for wing ISH).(114)Rinse once and wash 3× with PBT.(115)Perform a second fixation of the tissues for 30 min in 1 mL of fixation buffer at room temperature. Note that the fixation buffers for wing and abdomen ISH are different; therefore, see Section 6 for fixation buffer preparation.(116)Rinse once and wash 5× with PBT.(117)Replace the last PBT wash with 1 mL of the Proteinase K solution.


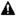
**CRITICAL STEP** Freshly prepare the Proteinase K solution on ice and use it within the same hour or two. For abdominal ISH, dilute 1 µL of Proteinase K stock [10 mg/mL in PBS] in 99 µL of PBT. Take 4 µL of this dilution and add it to 1 mL of PBT. For wing ISH, mix 0.4 µL of Proteinase K [10 mg/mL in PBS] with 1 mL of PBT.

(118)Incubate the tissues at room temperature for 10 min.


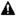
**CRITICAL STEP** Incubate the wings and abdomens in Proteinase K for 10 min for pupal stages P6 to P10 and 20 min for P11 to P15 to increase the tissue permeability and reduce background staining.

(119)Rinse 2× with PBT.(120)Wash 2× with PBT.(121)Post-fix the tissues for 30 min in 1 mL of fixation buffer at room temperature.(122)Wash 5× with PBT.(123)Wash in 1:1 PBT:hybridization solution (*v/v*).(124)Wash 3× with hybridization solution at room temperature.(125)Pipette the tissues into a 2 mL Eppendorf tube with a cut 1 mL tip.(126)Set the heating block to 80 °C.(127)Prehybridize the tissues in 500 µL of the hybridization solution and incubate at 65 °C for 1 h.(128)Dilute the probe (1:500) by adding 1 µL of the probe to 500 µL of hybridization solution in a 2 mL Eppendorf tube.(129)Incubate the diluted probe for 5 min in the heating block at 80 °C.(130)Immediately put the probe on ice to prevent secondary RNA structure formation.(131)Remove as much of the hybridization solution in (Step 127) without damaging the tissues and replace with the diluted probe on ice.(132)Incubate at 65 °C for >18 h to a maximum of 3 d, and gently swirl every couple of hours (not necessary during the night).


**Day 2**


(133)Pre-heat 3 mL of hybridization solution per sample to 65 °C on a dry-heating block.(134)Transfer the samples back into a clean glass-viewing dish by pipetting, using a cut 1-mL tip for abdomens and a cut 200-µL tip for wings.(135)Rinse once with pre-heated hybridization solution.(136)Incubate in pre-heated hybridization solution at 65 °C for 1 h.(137)Wash 3× at 65 °C for 30 min in pre-heated hybridization solution.(138)Prepare 1.5 mL of 1:1 PBT:hybridization solution (*v/v*).(139)Wash 2× with 1:1 PBT: hybridization solution (*v/v*) at room temperature.(140)Wash 5× with PBT.(141)Pipette the tissues with the 5th PBT wash into a 2 mL Eppendorf tube, using a cut 1-mL tip for abdomens or a cut 200-µL tip for wings.(142)Remove most of the PBT (leave about 50 µL PBT in the tube) and place the tissues on ice.(143)On ice in a separate 2-mL Eppendorf tube, add 0.2 µL of the Roche α-DIG AP Fab fragments to 1200 µL of PBT to result in a 1:6000 dilution.(144)Add 300 µL of the 1:6000 diluted Roche α-DIG-AP F_ab_ fragments to each sample and incubate at 4 °C overnight.


**Day 3**


(145)Transfer the samples back into a clean glass-viewing dish by pipetting, using a cut 1-mL tip for abdomens and a cut 200-µL tip for wings.(146)Wash 5× with PBT at room temperature.(147)Wash 3× with staining buffer at room temperature (see Section 6).(148)Remove the epidermal tissue layer from the puparium (outer tan shell). This step is required for abdominal ISH only.(149)Prepare the staining solution: add 2.8 µL of NBT (50 mg/mL) and 1.4 µL of BCIP (50 mg/mL) to 400 µL of staining buffer. Mix and keep in the dark.(150)Replace the last wash with 400 µL of staining solution.(151)Incubate in the dark at room temperature.(152)Check for signal development (purple stain) every 20 min.(153)Stop staining after the expression patterns look good, rinse once, and wash 2× with staining buffer.(154)Rinse once and wash 2× with PBT.(155)The tissues are now ready to be imaged. Use the information in (Appendix C) to image the tissues.

### 3.13. Timing

Probe making

Steps 1–4, A-tail genomic PCR: 4 hSteps 5–13, Gel extraction and PCR product purification: 30 minSteps 15–16, Ligation: 18 h (overnight)Steps 17–28, Transformation of *DH5-α* cells and colony PCR: 24 h (overnight)Steps 29–32, Culturing the positive colonies: 18 h (overnight)Steps 33–43, Plasmid extraction from a positive clone (mini-prep): 40 minSteps 44–47, Insertion direction PCR: 3 hSteps 48–54, RNA probe synthesis: 5 hSteps 55–64, RNA probe quality check and precipitation: 2 h

Dissection and ISH

Steps 65–84, *Drosophila* pupa collection and processing: 4 h (This excludes the time for larva collection and the periods of pupal development)Steps 85–105, *Drosophila* pupa processing for wing ISH: 4 hSteps 106–155, *Drosophila* abdominal or wing ISH: 3 d

## 4. Expected Results

The images of the gene expression patterns on *Drosophila* wings and abdomens generated using this protocol are shown in Figure 4, Figure 5, Figure 6 and Figure 7. This protocol has been used to generate quality ISH images for toolkit genes during early pupal stages on wings and abdomens of non-model *Drosophila* species [6,20] (Figure 4 and Figure 5).

The pupal dissection steps determine the final orientation of the pupa, whereby improper dissection can lead to the loss of important features, which might affect the image quality and interpretation. Through pupa abdominal ISH, we have shown that the *yellow* gene is expressed in all six rows of spots foreshadowing the adult *D. guttifera* and *D. quinaria* patterns (Figure 6).

It may be technically challenging to use one ISH image to show gene expression patterns in all the six rows of spots on the abdomen of some *Drosophila* species in the *quinaria* species group. Therefore, the mastery of the anatomy of the pupa and proficiency in making different types of cuts are very important to ensure that all spot rows are revealed in at least two ISH images. For example, the mid-cut pupa used for abdominal ISH shows the lateral, median, and/or dorsal rows of spots, which prefigure the spots in the lateral region of the adult fly’s abdomen (Figure 6a and Figure 7c), while the lateral cut shows the dorsal and median rows of spots foreshadowing the spots in the dorsal region of the adult fly’s abdomen (Figure 6b and Figure 7a). However, the lateral cut may occasionally reveal all six rows of spots as shown in (Figure 7b). Interestingly, we have shown this protocol to work in a wide range of rarely studied fruit flies in the *quinaria* species group, such as *D. deflecta*, *D. guttifera*, *D. recens*, *D. quinaria*, *D. subpalustris*, and *D. palustris.* This protocol contains the necessary information to facilitate the study of novel gene expression patterns in rare and unstudied fruit flies in the future.

## 5. Troubleshooting

This protocol is accompanied by an important troubleshooting table that summarizes a variety of problems that we have encountered in the past and offers appropriate solutions (Table 12).

## 6. Reagents Setup

*20× TAE.* Add 1600 mL distilled water and a stir bar to a 2 L beaker, add 193.6 g of Trizma Base, 46 mL Glacial Acetic Acid, 1.5 g EDTA. Stir until everything dissolves. Adjust the pH to 8.0 with about 26 mL of 37% HCl. Add distilled water to make 2 L and stir a little more.*LB medium*. Add 1800 mL of distilled water and a stir bar to a 2 L beaker. Add 20 g of Tryptone, 10 g Yeast extract, and 20 g NaCl. Set pH to 7.5 with about 700 µL 5 M NaOH. Adjust volume with distilled water to 2 L and stir a little more. Fill into 1 L bottles with about 600 mL, then autoclave.500 mL *Ampicillin agar plates*. Add 7.5 g of agar into 500 mL LB medium in a 1 L bottle and autoclave. Let it cool down until you can touch the bottle with your hands. Add 1 mL of ampicillin (at 100 mg/mL) per liter of agar to obtain a final concentration of 200 µg/mL. Swirl the medium to mix before pouring; be careful not to introduce bubbles. Pour into the medium-sized Petri dish, until it covers the bottom, approximately 30 mL. Store at 4 °C for 3 weeks.*Hybridization solution*. Add 200 mL of formamide and 100 mL 20× SSC. Set pH to 5.5 (check with color strips) and filter-sterilize. Then, add 4 mL of salmon sperm DNA (10 mg/mL), 40 mg heparin, 400 µL Tween 20, and 96 mL H_2_O. Mix thoroughly and store at −20 °C.*10× PBS*. Add 1800 mL of distilled water and a stir bar to a 2-L beaker. Add 21.4 g of K_2_HPO_4_, 10.3 g KH_2_PO_4_, and 163.6 g NaCl. Adjust volume to 2 L and stir a little more. Fill into 1-L bottles with about 600 mL, then autoclave. Note: The pH is at around 6.5 now. Prepare 1× PBS by making 1:10 dilution in distilled water. This will cause the pH to go up to 7.0–7.2. This is where the pH should be.*PBT.* Add 100 mL of 10× PBS, 1 mL Triton X-100, and 900 mL distilled H_2_O.*20× SSC.* Add 700 mL of distilled water and a stir bar to a 1-L beaker. Add 175.3 g of NaCl and 88.2 g sodium citrate. Adjust pH to 7.0 with about 2 drops of 37% HCl. Adjust the volume with distilled water to 1 L and stir. Fill into two 1-L bottles, then autoclave.*Staining buffer* (50 mL). Add 1 mL of 5 M NaCl, 2.5 mL 1 M MgCl_2_, 2.5 mL 2 M Tris pH 9.5, 50 µL Tween 20, and 44 mL distilled H_2_O. Make fresh staining buffer each time you perform an *in situ* and mix thoroughly before use.*Staining solution* (400 µL). Add 2.8 µL of NBT (50 mg/mL) and 1.4 µL BCIP (50 mg/mL) into 400 µL of freshly prepared staining buffer. Make fresh staining solution when required. Keep in the dark.*Fixation buffer for abdominal ISH* (40 mL). To a clean 50-mL Falcon tube, add 4 mL of 10× PBS, 85 µL 5 M NaOH, 20 mL sterile distilled H_2_O, and 80 mg deoxycholic acid. Vortex until the milky color (undissolved deoxycholic acid) disappears, then add 10 mL of 16% paraformaldehyde. Fill up to 40 mL with sterile distilled H_2_O. Store at 4 °C for maximum of one-month.*Fixation buffer for wings ISH (PBT + 4% PFA)*. To a clean 50-mL Falcon tube, add 4 mL of 10× PBS, 10 mL 16% paraformaldehyde, 400 µL Triton X-100, and add 25.6 mL sterile distilled H_2_O. Store at 4 °C for maximum of one month.

## Figures and Tables

**Figure 1 mps-04-00020-f001:**
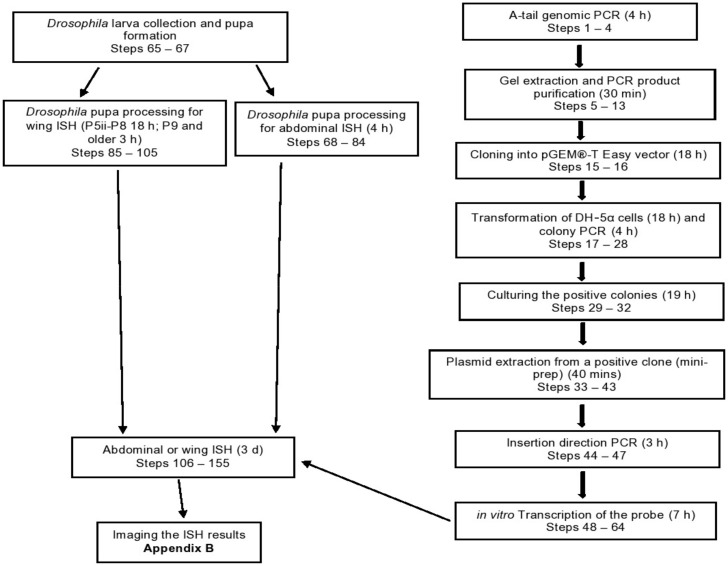
Overview of the ISH procedure in the wings and abdomens of *Drosophila* species. The major steps are shown sequentially in the boxes linked by arrows. The duration of each step is indicated in parentheses inside the boxes. Boxes on the right side describe the probe preparation steps, and boxes on the left side illustrate the sample preparation and ISH steps.

**Figure 2 mps-04-00020-f002:**
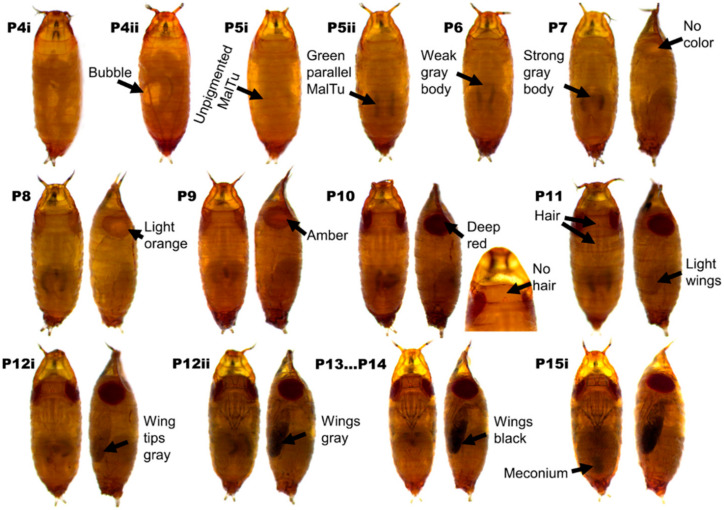
*D. guttifera* pupal developmental stages are labeled as described by Bainbridge and Bownes [25] for *D. melanogaster* and adopted for *D. guttifera* by Werner et al. [6]. (MaITu stands for Malpighian tubule).

**Figure 3 mps-04-00020-f003:**
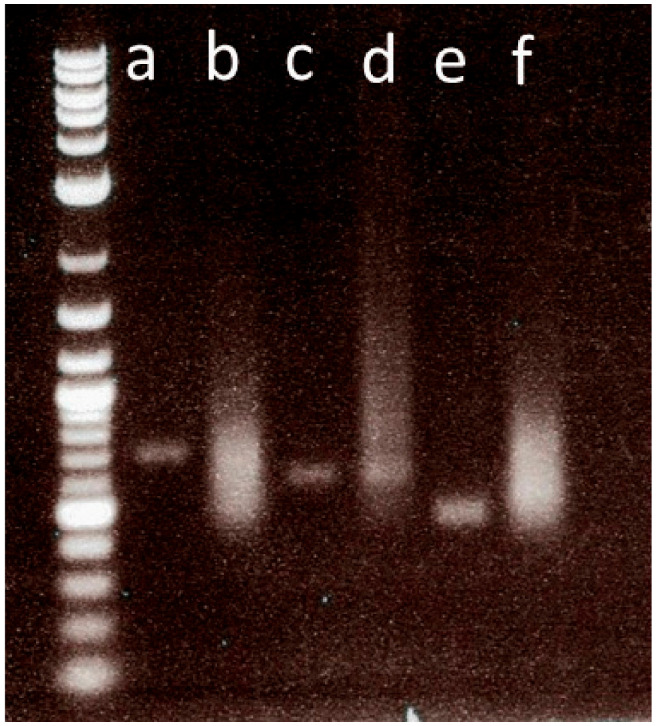
Probe quantity determination. An agarose gel image showing the quantity of anti-sense RNA probes. (**a**,**c**,**e**) represent PCR templates for the probes’ synthesis. (**b**,**f**) High-quantity probes showing thick probe smears (**d**) Low-quantity probe.

**Figure 4 mps-04-00020-f004:**
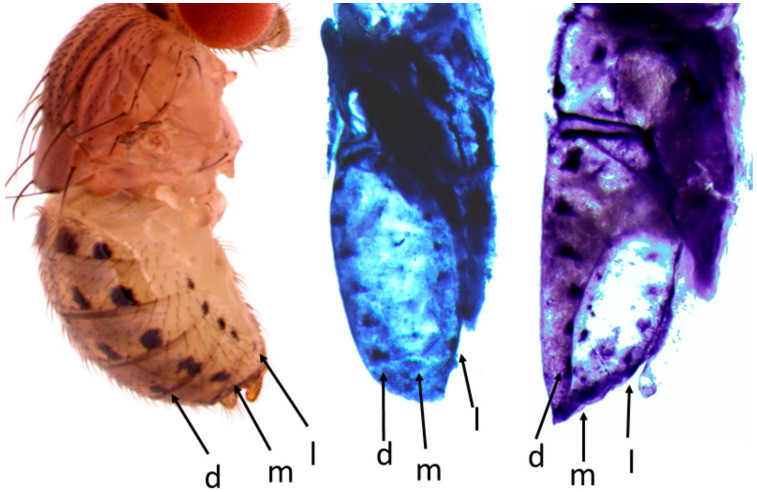
The mRNA expression pattern of *wingless* (*wg*) in the early pupal stage (P7) of *D. guttifera* foreshadowing the adult abdominal spot pattern. The spot rows are labeled as lateral (l), median (m), and dorsal (d).

**Figure 5 mps-04-00020-f005:**
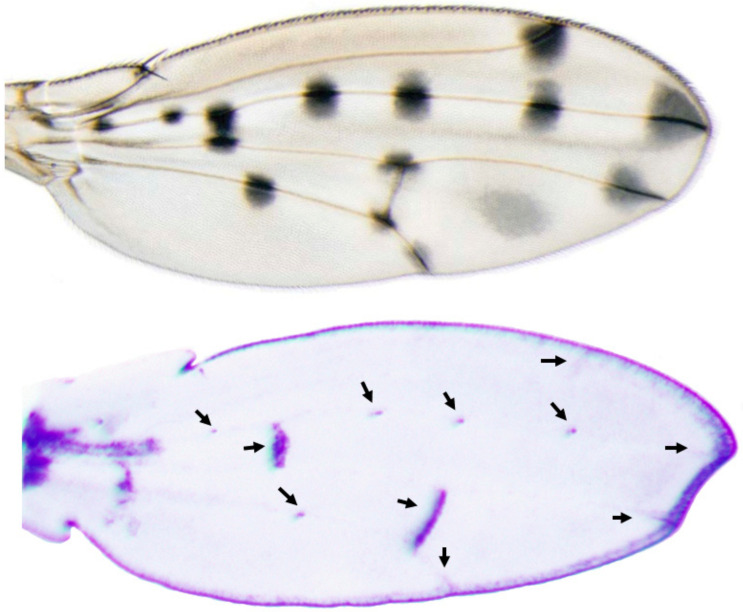
The *wingless* mRNA expression foreshadows the adult spot pattern on the wing of *D. guttifera*. This image has been previously published in Werner et al. [6].

**Figure 6 mps-04-00020-f006:**
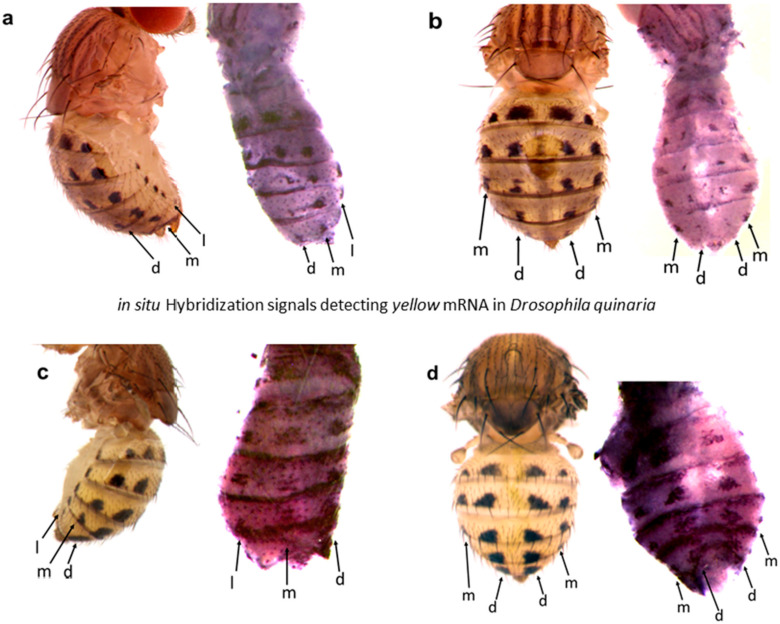
*in situ* Hybridization signals detecting *yellow* mRNA during pupal stage P10 of *D. guttifera* and *D. quinaria* foreshadow the abdominal spot patterns of the adults. (**a**,**c**) Dorsal cuts showing the lateral pattern of *yellow* mRNA expression in *D. guttifera* and *D. quinaria*, respectively. (**b**,**d**) Lateral cuts showing the dorsal pattern of *yellow* mRNA expression in *D. guttifera* and *D. quinaria*, respectively. The spot rows are designated as lateral (l), median (m), and dorsal (d).

**Figure 7 mps-04-00020-f007:**
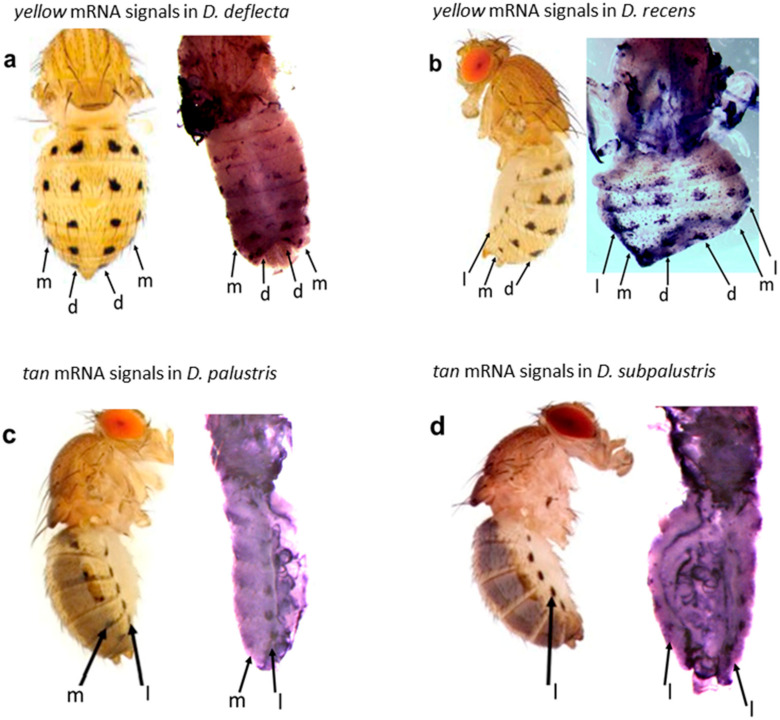
The *in situ* hybridization signals of the *yellow* (*y*) and *tan* (*t*) transcripts during *D. deflecta*, *D. recens*, *D. palustris*, and *D. subpalustris* pupal development prefigure the adult abdominal spot patterns. The images of the adult flies have been previously published in Werner et al. [23]. (**a**,**b**) *yellow* mRNA expression in *D. deflecta* and *D. recens* respectively, during pupal stage P10. (**c**,**d**) *tan* mRNA expression in *D. palustris* and *D. subpalustris*, respectively, during pupal stage P11, as shown in Dion et al. [5].

**Table 1 mps-04-00020-t001:** MeanGreen PCR reaction mix.

Reagent	Volume Per Reaction (µL)
Taq 2× MeanGreen Master Mix	12.5
Forward primer (10 pmol/µL)	1.25
Reverse primer (10 pmol/µL)	1.25
Genomic DNA (0.09–0.2 µg/µL)	0.25
d H_2_O	9.8
Total	25.0

**Table 2 mps-04-00020-t002:** Cycling conditions for genomic DNA amplification.

Reaction	Temperature	Time	Cycle Number
Initial Denaturation	95 °C	5 min	35 cycles
Denaturation	92 °C	30 s	
Annealing	X °C	30 s	
Extension	72 °C	1 min per kb	
Final Extension	72 °C	5 min	

**Table 3 mps-04-00020-t003:** Reaction mix for adding A-tails to the PCR products.

Reagent	Volume Per Reaction (µL)
PCR product (0.05–0.1 µg/µL)	7.0
dATP/dNTP (10 mM)	1.0
10× PCR buffer	1.0
Taq Polymerase (5 U/ µL)	1.0
Total	10

**Table 4 mps-04-00020-t004:** Reaction mix for ligation of A-tailed PCR product with the pGEM^®^-T Easy vector.

Reagent	Volume Per Reaction (µL)
A-tailed PCR product (0.05–0.1 µg/µL)	3.5
2× ligation buffer	5
pGEM^®^-TEasy vector (0.05 µg)	0.5
T4 DNA ligase	1
Total	10

**Table 5 mps-04-00020-t005:** Reaction mix for colony PCR.

Reagent	Volume Per Reaction (µL)
10× PCR buffer	1
dNTP mix (10 mM)	0.5
Internal forward primer (10 pmol/µL)	0.5
Internal reverse primer (10 pmol/µL)	0.5
Taq polymerase (5 U/µL)	0.2
d H_2_O	15.3
Bacterial suspension	2
Total	20

**Table 6 mps-04-00020-t006:** Cycling condition for colony PCR.

Reaction	Temperature	Time	Cycle Number
Initial Denaturation	95 °C	5 min	35 cycles
Denaturation	92 °C	30 s	
Annealing	X °C	30 s	
Extension	72 °C	1 min per kb	
Final Extension	72 °C	10 min	

**Table 7 mps-04-00020-t007:** Reaction mix for insertion direction PCR.

Reagent	Volume Per Reaction (µL)
10× PCR buffer	2.0
dNTP mix (10 mM)	0.5
M13F (vector primer) (10 pmol/µL)	0.5
Internal gene-specific primer (forward or reverse) (10 pmol/µL)	0.5
Taq polymerase (5 U/µL)	0.2
dH_2_O	16.2
Mini-prep DNA (insert in pGEM^®^-TEasy)	0.1
Total	20

**Table 8 mps-04-00020-t008:** Cycling condition for insertion direction PCR.

Reaction	Temperature	Time	Cycle Number
Initial Denaturation	95 °C	5 min	35 cycles
Denaturation	92 °C	30 s	
Annealing	45 °C	30 s	
Extension	72 °C	1 min per kb	
Final Extension	72 °C	5 min	

**Table 9 mps-04-00020-t009:** Reaction mix for PCR amplification of the cloned insert.

Reagent	Volume Per Reaction (µL)
Taq 2× MeanGreen Master Mix	12.5
M13F (vector forward primer) (10 pmol/µL)	1.25
M13R (vector reverse primer) (10 pmol/µL)	1.25
Mini-prep DNA (0.1–0.5 µg/µL)	0.1
d H_2_O	9.9
Total	25

**Table 10 mps-04-00020-t010:** Reaction mix for anti-sense probe preparation.

Reagent	Volume Per Reaction (µL)
Purified PCR product (0.05–0.1 µg/µL)	6.5
10× NTP labeling mixture	1
10× Transcription buffer	1
Protector RNAse inhibitor	0.5
Sp6 RNA polymerase or T7 RNA polymerase	1
Total	10

**Table 11 mps-04-00020-t011:** Reaction mix for probe precipitation.

Reagent	Volume Per Reaction (µL)
Probe (from step 54)	9
Linear acrylamide (5 µg/µL)	1
3 M Sodium acetate pH 5.5	1
200-Proof ethanol	22.5
Total	33.5

**Table 12 mps-04-00020-t012:** Troubleshooting guide. A table summarizing the problems that we have encountered in the past with appropriate actions to solve each problem.

Step	Problem	Possible Cause	Solution
3	No PCR product	Poor primer design	Ensure the primers are made from the highly conserved region. Check the primer design
		Low-quality genomic DNA	Check the genomic DNA on agarose gel
		Poor cycling conditions	Adjust the PCR conditions as needed
26	No positive colonies	Failed ligation	Ensure A-tail are added to the PCR products before ligation with the pGEM^®^-TEasy vector. Note that the A-tails may fall off by keeping the PCR product at −20 °C for one month
47	Lack of PCR product after insertion direction PCR	No insert or wrong insert in the pGEM^®^-TEasy vector	Perform a PCR with the internal primers to check that the correct insert is in the vector or sequence the plasmid instead.
		Wrong primer pairs are used	Ensure to use the internal forward versus M13F, and internal reverse versus M13F primer pairs
		The species-specific internal primers’ melting temperature does not match the M13F vector primer	Design species-specific internal primers with melting temperatures between 45 °C and 50 °C
58	Agarose gel analysis showing no RNA probe	Poor probe synthesis reaction	Repeat probe synthesis reaction. Avoid RNAse interference
133	Purple staining is visible but weak	Low gene expression or inadequate staining time	Stain the tissue longer (overnight). Ensure the tissues are adequately permeabilized with the Proteinase K solution
	High background staining	Over-staining	Check for signal every 30 min
		Poor tissue treatment	Use the appropriate Proteinase K concentration, and ensure correct duration of treatment
	No *in situ* signal	Gene may not be expressed	Always perform positive and negative controls to rule out bad reagents
		The traditional negative controls for an antisense probe are the sense probe. Therefore, the use of a sense probe will give no signal.	Use the appropriate RNA polymerase to synthesize the anti-sense probe Be sure to use an antisense probe for your test samples
		RNAse might have destroyed the mRNA or the RNA probe	Avoid talking, coughing, and sneezing into the sample. Always use gloves when performing ISH

## Data Availability

Not applicable.

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
