# Peer review of "RNA In Situ Hybridization for Detecting Gene Expression Patterns in the Abdomens and Wings of Drosophila Species"

_mps, 2021, doi:10.3390/mps4010020_

Round 1
Reviewer 1 Report
In this manuscript, the authors present detailed protocol for mRNA in situ hybridization (ISH) in developing pupal wings and abdomens in some Drosophila species. It seems this manuscript for the protocol is well organized and clearly written. As the authors mentioned in introduction, no detailed protocol to perform RNA ISH for pupal wings and abdomens in Drosophila species has been published so far. Therefore, I agree with the author’s claim that this protocol will facilitate the study of novel gene expression patterns in rare and unstudied fruit flies, and I believe this manuscript is worthy of publication in Methods and Protocols. I have only few minor suggestions regarding the text.
L33: It would be better to add “(anti-DIG-AP)” behind alkaline phosphatase.
L111: change “an” to “and”
L145, “The carcasses were then fixed”: Which fixative solution are the authors using in this case? Paraformaldehyde?
L432: How much speed are the authors using for this spin?
L493: It would be better to add “for abdominal ISH” behind fixation buffer.
Appendix D: Why did the authors describe the processing for pupal wings before ISH in Appendix? I think this protocol for the pupa processing is one of important parts for this manuscript. Thus I suggest to move this protocol described in Appendix D just before 3.11. ISH of Drosophila abdomens and wings (L503).
Author Response
In this manuscript, the authors present detailed protocol for mRNA in situ hybridization (ISH) in developing pupal wings and abdomens in some Drosophila species. It seems this manuscript for the protocol is well organized and clearly written. As the authors mentioned in introduction, no detailed protocol to perform RNA ISH for pupal wings and abdomens in Drosophila species has been published so far. Therefore, I agree with the author’s claim that this protocol will facilitate the study of novel gene expression patterns in rare and unstudied fruit flies, and I believe this manuscript is worthy of publication in Methods and Protocols. I have only few minor suggestions regarding the text.
L33: It would be better to add “(anti-DIG-AP)” behind alkaline phosphatase.
Response: We have made the change.
L111: change “an” to “and”
Response: We have changed the text.
L145, “The carcasses were then fixed”: Which fixative solution are the authors using in this case? Paraformaldehyde?
Response: Yes, we use PBT + 4 % PFA. The preparation of the fixation solution is explained under section 6.0 (Reagents setup)
L432: How much speed are the authors using for this spin?
Response: The speed is 15000 rpm. We have added the revolution speed to the text.
L493: It would be better to add “for abdominal ISH” behind fixation buffer.
Response: We have added it to the text.
Appendix D: Why did the authors describe the processing for pupal wings before ISH in Appendix? I think this protocol for the pupa processing is one of important parts for this manuscript. Thus I suggest to move this protocol described in Appendix D just before 3.11. ISH of Drosophila abdomens and wings (L503).
Response: We have deleted Appendix D and moved it to Section 3.11 as suggested.

Reviewer 2 Report
The manuscript entitled "RNA in situ hybridization for detecting gene expression patterns 2 in the abdomens and wings of Drosophila species" by Shittu et al., Describes in detail a protocol to perform in situ hybridization in pupal wings and abdomens that can be applied to different species of Drosophila.
The protocol is described in detail and is easy to follow, so it can be said that the methodology can be reproduced by other laboratories. They precisely indicate the critical points as well as the possible problems that may arise during the protocol and how to solve them.
This protocol can be very useful for groups working with different Drosophila species and can even be the basis for in situ hybridizations with other diptera in the pupal stage. Therefore I consider that it covers the merits of being published in MP. I only have two points that will improve the presentation of this work.
1.- The table in section 3.9 is not well represented and it has to be corrected.
2.- In all the figures in which the hybridizations are shown, it is important to show a negative control, in other words similar organisms hybridized with the sense probe. Since it is a protocol report, showing negative controls is important
Author Response
Open Review
Reviewer 2
Open Review
(x) I would not like to sign my review report
( ) I would like to sign my review report
English language and style
( ) Extensive editing of English language and style required
( ) Moderate English changes required
( ) English language and style are fine/minor spell check required
(x) I don't feel qualified to judge about the English language and style
|
Yes |
Can be improved |
Must be improved |
Not applicable |
|
|
Does the introduction provide sufficient background and include all relevant references? |
(x) |
( ) |
( ) |
( ) |
|
Is the research design appropriate? |
(x) |
( ) |
( ) |
( ) |
|
Are the methods adequately described? |
(x) |
( ) |
( ) |
( ) |
|
Are the results clearly presented? |
( ) |
(x) |
( ) |
( ) |
|
Are the conclusions supported by the results? |
(x) |
( ) |
( ) |
( ) |
Comments and Suggestions for Authors
The manuscript entitled "RNA in situ hybridization for detecting gene expression patterns 2 in the abdomens and wings of Drosophila species" by Shittu et al., Describes in detail a protocol to perform in situ hybridization in pupal wings and abdomens that can be applied to different species of Drosophila.
The protocol is described in detail and is easy to follow, so it can be said that the methodology can be reproduced by other laboratories. They precisely indicate the critical points as well as the possible problems that may arise during the protocol and how to solve them.
This protocol can be very useful for groups working with different Drosophila species and can even be the basis for in situ hybridizations with other diptera in the pupal stage. Therefore I consider that it covers the merits of being published in MP. I only have two points that will improve the presentation of this work.
1.- The table in section 3.9 is not well represented and it has to be corrected.
Response: We did not find any problem with the table in section 3.9.
2.- In all the figures in which the hybridizations are shown, it is important to show a negative control, in other words similar organisms hybridized with the sense probe. Since it is a protocol report, showing negative controls is important
Response: Traditionally, people did a sense control, but we stopped using the sense probe since it was discovered that there can be micro RNAs that can bind to the sense probe. There is no perfect control; however, we use genes that are not expressed as negative controls, and the genes that are expressed as positive controls (e.g., the yellow gene at P10 in D. guttifera). We did not take pictures of our negative controls.

Reviewer 3 Report
This is a well-written methods paper that I have read in detail. I have vast experience in whole mount DIG-labeled in situs on Drosophila embryos and dissected ovaries using DNA, anchored PCR-generated and RNA probes using a variety of hybridization conditions. The protocol is straightforward and resembles in detail many of the protocols that I have used in the embryo and the ovary, which are easy and hard tissues to get good in situ results, respectively.
These detailed approaches and excellent figures (note that permissions for one figure may be required for publication here) will encourage a researcher like me, who is competent to dissect third instar larva, but is reluctant to dissect pupa, the encouragement to do so. I appreciate the way that the paper is written in an orderly style, with common methods, such as probe generation, presented followed by variations in methods based on tissue applications interspersed in a linear presentation. Below I detail a few places where the procedure could be clarified or supplemented:
Line 335 specify “fresh” small pipet tip (as it reads, one might use the same tip and cross-contaminate)
Line 539 1.5 ml eppendorfs are more common – could these be used or is 2.0ml required?
Line 114 volumes of washes are needed here and throughout
Line 119 volumes and times of washes are needed throughout
Line 126 unclear if this dissection is the pupa case or an epithelial layer associated with the abdomen – please add some details for this step – again a non-expert in pupa like me would be helped by a detailed description of this step
Page 22 Troubleshooting step 26 – alternatives to T/A cloning would be helpful, notably topo-TA cloning which can be more efficient for researchers stumbling on a recalcitrant ligation step
Page 23 step 47 offer sequencing as an alternative to determine orientation - my students go to it first, and frankly it would be cheaper and more reliable than internal primers that must be ordered and shipped
Author Response
These detailed approaches and excellent figures (note that permissions for one figure may be required for publication here) will encourage a researcher like me, who is competent to dissect third instar larva, but is reluctant to dissect pupa, the encouragement to do so. I appreciate the way that the paper is written in an orderly style, with common methods, such as probe generation, presented followed by variations in methods based on tissue applications interspersed in a linear presentation. Below I detail a few places where the procedure could be clarified or supplemented:
Line 335 specify “fresh” small pipet tip (as it reads, one might use the same tip and cross-contaminate)
Response: We have made the change.
Line 539 1.5 ml eppendorfs are more common – could these be used or is 2.0ml required?
Response: We prefer to use 2.0 mL Eppendorf tubes due to its round and wide bottom unlike the narrow 1.5 mL bottom. The tissues are well spread out in a 2.0 mL tube.
Line 114 volumes of washes are needed here and throughout
Response: The instructions about volumes and times are stated in Lines 453-460 (see the CRITICAL STEP under Section 3.10)
Line 119 volumes and times of washes are needed throughout
Response: The instructions about volumes and times are stated in Lines 453-460 (see the CRITICAL STEP under Section 3.10)
Line 126 unclear if this dissection is the pupa case or an epithelial layer associated with the abdomen – please add some details for this step – again a non-expert in pupa like me would be helped by a detailed description of this step
Response: The whole pupa will be dissected as described under section 3.10. The puparium is needed to provide support for the tissue throughout the ISH process. However, the tissue should be separated from the puparium before staining.
Page 22 Troubleshooting step 26 – alternatives to T/A cloning would be helpful, notably topo-TA cloning which can be more efficient for researchers stumbling on a recalcitrant ligation step
Response: People can clone however they like. We have presented in this article how to easily clone into the pGEM®-T Easy vector because T/A cloning into this vector is very basic, simple, and we highly recommend it. This vector also has all the primer sites and promoters lined up in the correct order. Other vectors may be differently aligned, and the sense versus antisense issue would have to be reevaluated.
Page 23 step 47 offer sequencing as an alternative to determine orientation - my students go to it first, and frankly it would be cheaper and more reliable than internal primers that must be ordered and shipped
Response: This observation is correct. We have included sequencing as an option.

Reviewer 4 Report
Shittu and colleagues have assembled a thorough description of the processes and workflows required to set up an in situ hybridisation protocol for drosophilid pupae. While, in essence, very little of these procedures is new, it should be very useful to the researcher that wanted to perform these experiments without previous experience, with some very useful indications specific for the handling of pupa in the context of this type of experiment. Their thoroughness in this regard is very useful and therefore I recommend publication. I have a one request and a few suggestions that are perhaps a matter of opinion but that in my view may improve the manuscript. In page 2, lines 47-50 the authos say: "Although several studies have used the RNA ISH technique to detect and characterize gene expression patterns on pupal wings and abdomens of various Drosophila species [6,7,17,18], no protocol describing this technique in Drosophila pupae has been published in any scholarly journal." This may be true in the form of fully detailed protocol, but being that many of the steps described here are relatively generic molecular biology procedures, I think it should be acknowledged the work and methods described in Ben J. Vincent et al., "An Atlas of Transcription Factors Expressed in Male Pupal Terminalia of Drosophila melanogaster" G3: Genes, Genomes, Genetics December 1, 2019 vol. 9 no. 12 3961-3972. At least in the list of references of that paragraph. As for the suggested changes: - In Figure 2, some of the indicated anatomical landmarks are not clearly seen in the photos - this is usual in photo guides, and a reason why field guides usually are based on drawings and diagrams. I find particularly difficult to see the White Malpighian tubules, and the Weak vs Strong gray body. Being that this protocol will be most useful to the inexperienced experimenters and its value being based mostly on being a one-stop shop for this experiment, why not including a few diagrams, even if only in the less clear landmarks (e.g. like a replica of the photo, or an inset, in diagram form). This is done extensively in the cited references (25 and 26). - In page 5, describing the probe design, it is not very clear why a PCR primer pair is designed per species ("species-specific PCR primers") if later it is said that care has been made to ensure the amplicon has the same size in all species. It takes some reading to work out that the same amplicon is used for all the different species. Also, I think this should be justified, as while the authors say that they choose a well conserved area, the levels of identity are not described nor the impact that mismatches in the sequence may have on hybridisation efficiency is discussed. - In page 16 "Store the Petri dish in a species-specific moist chamber to prevent the larvae from drying out." This is a very minor thing but the wording suggests that each species requires a specific design of a humidity chamber... I guess this refers to labelling them carefully so specimens of different species are not mixed? Could this be worded a bit more directly? - It may be more interesting to show the expression patterns of different species together, e.g. fusing Figs 6 and 7 labelling in the figure which species is in each row (and that the probe is exactly the same). Similar labelling may be useful in fig 8, so one can quickly interpret the figure without reading all details in the legend. - Finally, in page 23 (troubleshooting no in situ signal) - perhaps mention specifically that the natural negative controls for an antisense probe is the sense probe? This isn't discussed anywhere in the protocol (that I could find) and for a protocol that explains even the basics of PCR amplification and TA-ligation, I think it woud be useful to mention this too.Author Response
Shittu and colleagues have assembled a thorough description of the processes and workflows required to set up an in situ hybridisation protocol for drosophilid pupae. While, in essence, very little of these procedures is new, it should be very useful to the researcher that wanted to perform these experiments without previous experience, with some very useful indications specific for the handling of pupa in the context of this type of experiment. Their thoroughness in this regard is very useful and therefore I recommend publication. I have a one request and a few suggestions that are perhaps a matter of opinion but that in my view may improve the manuscript. In page 2, lines 47-50 the authos say: "Although several studies have used the RNA ISH technique to detect and characterize gene expression patterns on pupal wings and abdomens of various Drosophila species [6,7,17,18], no protocol describing this technique in Drosophila pupae has been published in any scholarly journal." This may be true in the form of fully detailed protocol, but being that many of the steps described here are relatively generic molecular biology procedures, I think it should be acknowledged the work and methods described in Ben J. Vincent et al., "An Atlas of Transcription Factors Expressed in Male Pupal Terminalia of Drosophila melanogaster" G3: Genes, Genomes, Genetics December 1, 2019 vol. 9 no. 12 3961-3972. At least in the list of references of that paragraph.
Response: We have cited the article as suggested and added it to my list of references.
As for the suggested changes:
- In Figure 2, some of the indicated anatomical landmarks are not clearly seen in the photos - this is usual in photo guides, and a reason why field guides usually are based on drawings and diagrams. I find particularly difficult to see the White Malpighian tubules, and the Weak vs Strong gray body. Being that this protocol will be most useful to the inexperienced experimenters and its value being based mostly on being a one-stop shop for this experiment, why not including a few diagrams, even if only in the less clear landmarks (e.g. like a replica of the photo, or an inset, in diagram form). This is done extensively in the cited references (25 and 26).
Response: The “White Malpighian tubules” are not actually white in color when viewed under the microscope; they are unpigmented. These unpigmented Malpighian tubules are very hard to photograph but they can be seen in the microscope. We have changed the label to unpigmented Malpighian tubules.
- In page 5, describing the probe design, it is not very clear why a PCR primer pair is designed per species ("species-specific PCR primers") if later it is said that care has been made to ensure the amplicon has the same size in all species. It takes some reading to work out that the same amplicon is used for all the different species. Also, I think this should be justified, as while the authors say that they choose a well conserved area, the levels of identity are not described nor the impact that mismatches in the sequence may have on hybridisation efficiency is discussed.
Response: The probes that were used to perform ISH for the results shown in this article were made from the same amplicon. Thus, we have removed “species-specific” from the sentence. The statement now reads “the process starts with designing PCR primers to amplify a partial protein-coding region from a single exon, using the GenePalette software [25] as described in (Appendix A)”.
Also, from Line 118 to121 we added this statement; “It should be noted that the use of the same PCR primers that are designed from a highly conserved exon to amplify the amplicon in different Drosophila species may lead to mismatches in the sequence, which can reduce the hybridization efficiency”.
- In page 16 "Store the Petri dish in a species-specific moist chamber to prevent the larvae from drying out." This is a very minor thing but the wording suggests that each species requires a specific design of a humidity chamber... I guess this refers to labelling them carefully so specimens of different species are not mixed? Could this be worded a bit more directly?
Response: Here, we collect wandering third-instar larvae in a Petri dish with moist Kleenex paper on the bottom, then the Petri dish is finally stored in a moist chamber. The larvae are capable of moving from one Petri dish to another therefore getting mixed up. This is the reason we separate different species into different chambers. Although, some people collect pupae directly from the culture bottle, which is also fine and does not require separation. However, we prefer to check the development of the pupae every day in order to dissect them at the right time.
- It may be more interesting to show the expression patterns of different species together, e.g. fusing Figs 6 and 7 labelling in the figure which species is in each row (and that the probe is exactly the same). Similar labelling may be useful in fig 8, so one can quickly interpret the figure without reading all details in the legend.
Response: The figures have been fused as suggested.
- Finally, in page 23 (troubleshooting no in situ signal) - perhaps mention specifically that the traditional negative controls for an antisense probe is the sense probe? This isn't discussed anywhere in the protocol (that I could find) and for a protocol that explains even the basics of PCR amplification and TA-ligation, I think it woud be useful to mention this too.
Response: We have adjusted the statement in the troubleshooting section. The statement now reads “The traditional negative controls for an antisense probe are the sense probe. Therefore, the use of a sense probe will give no signal”.

Reviewer 5 Report
In this protocol entitled “RNA in situ hybridization for detecting gene expression patterns in the abdomens and wings of Drosophila species”, Shittu et al. present the technical details for probe preparation and in situ hybridization (ISH) of pupal abdomens and wings from Drosophilids. The protocol is generally well-written and easy to understand. Pupal ISH is technically difficult because of the required dissection and the soft tissues, and there are no published protocols addressing this developmental stage. The protocol thus fills a need and is generally applicable beyond the evo-devo community for any Drosophila lab wishing to perform ISH at pupal stages, particularly of epidermal structures.
Below I detail suggestions or points where the protocol could be made clearer.
- Materials List & Equipment/Supplies List: Can these be divided into two sub-lists each, one corresponding to materials/equipment needed for Probe Preparation and the other for Dissection/ISH? These are two distinct protocols within the general ISH protocol, and it would be clearer for a student to understand what they need for each part.
- Step 23: add reference to “Reagents Setup” section.
- Step 27: I wasn’t sure what “independently suspend” meant. Maybe reword to “…and suspend each colony in 10 uL of dH2O in an independent 1.5-mL Eppendorf tube”.
- Table 2: The “internal” and “external” primer designation should be introduced somewhere in the text. I didn’t know what this referred to until I got to the very end of the protocol in the Appendix A.
- In general, at the beginning of each protocol section, it might make sense to add a line of reagents/materials that need to be prepared in advance. For example for 3 Ligation, before ever starting this part of the protocol the student/researcher should have prepared the ampicillin plates, the IPTG and X-Gal solutions, and the LB medium.
- Line 350(PAUSE STEP): I think this refers to the plate from step 27, right? Or is this referring to a re-streak after having grown-up the positive colonies? Technically, the plate could be stored before growing-up the positive colonies as the protocol uses selection medium. You can’t really pause for 7 days between step 32 and 33, which is currently where this PAUSE STEP comes in the protocol.
- Protocol 3.6 (steps 35-42): This should probably somewhere explicitly mention the mini-prep kit where one can find the resuspension, lysis, neutralization, wash and elution buffers and the columns. I presume the authors are using a kit, but I can’t find which one listed in the protocol or in the materials/reagents lists.
- Line 440/PAUSE STEP after step 64: Please be explicit in how the RNA probe is stored long-term. Is it dissolved in hybridization buffer or as a pellet under EtOH to be dissolved when you are ready to use it? Should it be aliquoted so it is only thawed once before use, or is freeze-thaw cycling for periodic use over 2 years OK?
- Step 72 & 73: I think these are the same step. The point is to cut the pupa in half, either dorsally or longitudinally, using a razor blade, right? “Dissect” is confusing here, as you aren’t really dissecting rather just cutting the pupa in half, or bisecting the pupa (also in line 476). Technically, I also have a question here. Early pupa are under pressure and if we just cut them in half they will explode. Most of my students just squish the tissue at first and it is mostly useless. Do the authors take any steps to prevent that in this protocol, or is this not a concern when looking at abdominal epidermis?
- Step 76 & 77: Again, I think these are the same step. 77 just gives more detail in how to perform 76. Also, I think this point that the pupa needs to remain attached to the pupal case should be made explicit above in step 72.
- Step 83&84: I am not clear here how to proceed if I will perform ISH immediately versus if I will store the samples for later use. Do all samples get treated with the EtOH dehydration series? Or only those that will be stored? Where do I jump into the steps listed under 3.11 if I don’t store the samples? Also, I think that “dehydrate” is clearer than “equilibrate” the pupa halves.
- Step 87: Is this the first step that requires the gel slick coating, or is this also necessary for the steps above?
- Step 92: Would “rehydrate” be more accurate than “equilibrate” for this step? To avoid any confusion so the reader realizes the solutions are the opposite of step 83?
- Step 94: It might be clearer to specify to perform a second fixation of the tissues in this step, as they were already fixed once. Also, just to check, this fixation buffer is the same as the first fixation buffer?
- Step 97: Can the authors specify which samples/stages should only be incubated for 10 min. with Proteinase K, as opposed to wings and abdomens at P11-P15 which should be incubated for 20 min? Is it possible to incubate with Proteinase K for too long?
- Step 115-117: Is this meant to be done in an oven at 65C, since the samples are in a glass-viewing dish?
- Step 122 & 127: Authors need to add “in a separate Eppendorf tube” to be clear the reagents are mixed-up in a different tube than what the samples are in.
- Day 3/Step 124-125: Is this at RT or at 4C?
- Step 126: Is pupal case separation performed in the Eppi? Or are the samples transferred to a glass view-dish, separated under a dissecting microscope, and then transferred back into an Eppi for staining?
- Step 130: So is staining performed in an Eppi? Is signal development (presumably checked under a dissecting microscope) clear in an Eppi? Or should samples be in a glass view-dish?
- Step 133: The imaging protocol can be found in Appendix B, was not at first apparent from the phrasing here. Also, shouldn’t this CRITICAL STEP be part of Appendix B?
- Section 3.12 Timing: This is a useful summary, but can it be divided into two sections for Probe-preparation and Dissection/ISH with a time-estimate for that part? Also, technically for Steps 65-84 the flies have to be crossed/flipped and staged so the preparation for pupal collection takes longer than 4h.
- Figures 4-8: Can the authors include a sense control image for one or more of these figures? I think this actually should also be mentioned clearly in the probe synthesis and design sections as well, that controls are absolutely necessary to test staining specificity. I presume the authors use a sense control, but are there any other controls the authors regularly use and recommend for someone trying this protocol? For example, in Figure 7 can the authors show a control at P10 with a non-specific probe or without a probe to show the spot pattern is only evident in the ISH? In the legend for Figure 7 the authors should also mention that (A) and (B) are the dorsal and lateral cut view.
- Part 6. Reagents Setup, amp agar plates: Authors should include the step of pouring the amp + agar solution into 10 cm plates and letting them harden. Also proper storage of the plates until use.
- Appendix B: This imaging protocol should be clear in case the reader has never imaged ISH samples before. In step 1, what kind of well? In a glass view-dish or a depression microscope slide? Step 2: chances are people will not have the exact imaging set-up the authors use. What is critical about the objective and the camera specs to get appropriate images? Step 5/6: Can these steps also be performed with Image J/Fiji or with other image editing software? I do not have Helicon Focus or any Adobe software in my lab, for example, and we use Image J and Affinity products.
- Appendix C: I would reference the materials list as many people are probably not familiar with gel slick reagent.
- Appendix D: Step 5 – I presume the pupal case needs to be teased open before taking the pupa out by the head, and this should be added to the protocol. Do the authors also split the case a bit further? In my experience, the case needs to be about ½ open before I can pull a pupa out intact without just merely pulling off the head. Step 16 – It needs to be clear here how this protocol connects to steps 1-15 above, I presume step 16 comes after step 5, right? Also, can the authors please be more specific with “dissect pupal wings”? How do they do this? I can imagine using a surgical scissors to just cut-off the wing and a bit of the surrounding epidermis. Also for step 17, by “allow the wings to inflate” do the authors mean just incubate the dissected wings in buffer for 5-10 minutes until they have filled with buffer and inflated?
Author Response
In this protocol entitled “RNA in situ hybridization for detecting gene expression patterns in the abdomens and wings of Drosophila species”, Shittu et al. present the technical details for probe preparation and in situ hybridization (ISH) of pupal abdomens and wings from Drosophilids. The protocol is generally well-written and easy to understand. Pupal ISH is technically difficult because of the required dissection and the soft tissues, and there are no published protocols addressing this developmental stage. The protocol thus fills a need and is generally applicable beyond the evo-devo community for any Drosophila lab wishing to perform ISH at pupal stages, particularly of epidermal structures.
Below I detail suggestions or points where the protocol could be made clearer.
- Materials List & Equipment/Supplies List: Can these be divided into two sub-lists each, one corresponding to materials/equipment needed for Probe Preparation and the other for Dissection/ISH? These are two distinct protocols within the general ISH protocol, and it would be clearer for a student to understand what they need for each part.
Response: Some equipment/materials are important for both probe preparation and dissection/ISH. Also, if we did this separation, we would need to do separate list for the wings and abdomens ISH. Therefore, we made the list of equipment and reagent based on the journal’s requirement and formatting.
- Step 23: add reference to “Reagents Setup” section.
Response: Done, thank you.
- Step 27: I wasn’t sure what “independently suspend” meant. Maybe reword to “…and suspend each colony in 10 uL of dH2O in an independent 1.5-mL Eppendorf tube”.
Response: We reworded it as suggested.
- Table 2: The “internal” and “external” primer designation should be introduced somewhere in the text. I didn’t know what this referred to until I got to the very end of the protocol in the Appendix A.
Response: We have introduced the internal and external primers in the “Experimental Design” section.
- In general, at the beginning of each protocol section, it might make sense to add a line of reagents/materials that need to be prepared in advance. For example for 3 Ligation, before ever starting this part of the protocol the student/researcher should have prepared the ampicillin plates, the IPTG and X-Gal solutions, and the LB medium.
Response: It is a good idea; however, we expect the user to have read the protocol and prepare ahead of the experiment.
- Line 350(PAUSE STEP): I think this refers to the plate from step 27, right? Or is this referring to a re-streak after having grown-up the positive colonies? Technically, the plate could be stored before growing-up the positive colonies as the protocol uses selection medium. You can’t really pause for 7 days between step 32 and 33, which is currently where this PAUSE STEP comes in the protocol.
Response: The PAUSE STEP has been removed. We agree that the PAUSE STEP is not necessary.
- Protocol 3.6 (steps 35-42): This should probably somewhere explicitly mention the mini-prep kit where one can find the resuspension, lysis, neutralization, wash and elution buffers and the columns. I presume the authors are using a kit, but I can’t find which one listed in the protocol or in the materials/reagents lists.
Response: Plasmid mini-prep kit (Thermo Scientific, cat. no. K0503). This is mentioned in the “Materials” section Line 190. It has been added to the Section 3.6 topic.
- Line 440/PAUSE STEP after step 64: Please be explicit in how the RNA probe is stored long-term. Is it dissolved in hybridization buffer or as a pellet under EtOH to be dissolved when you are ready to use it? Should it be aliquoted so it is only thawed once before use, or is freeze-thaw cycling for periodic use over 2 years OK?
Response: The probe is dissolved in hybridization buffer, this is stated in Step 64. The probe should be aliquoted and stored. We have modified the sentence in the PAUSE STEP.
- Step 72 & 73: I think these are the same step. The point is to cut the pupa in half, either dorsally or longitudinally, using a razor blade, right? “Dissect” is confusing here, as you aren’t really dissecting rather just cutting the pupa in half, or bisecting the pupa (also in line 476). Technically, I also have a question here. Early pupa are under pressure and if we just cut them in half they will explode. Most of my students just squish the tissue at first and it is mostly useless. Do the authors take any steps to prevent that in this protocol, or is this not a concern when looking at abdominal epidermis?
Response: Steps 72 and 73 can be merged but we wanted to properly break things down for easy reading. “Dissect” has been replaced with “cut”. We recommend the type of razor we use in our lab (see the Materials section), ensure the pupae stick to the tape before performing a single rapid cut from the anterior to the posterior end of the pupae. Drag the razor, do not be tempted to press it down on the pupa while cutting. Cut and use at least 10 pupa halves for ISH. Your students should practice for at least one month to get comfortable before cutting for the experimental purpose.
- Step 76 & 77: Again, I think these are the same step. 77 just gives more detail in how to perform 76. Also, I think this point that the pupa needs to remain attached to the pupal case should be made explicit above in step 72.
Response: Steps 76 and 77 can be merged but we wanted to properly break things down for easy reading.
- Step 83&84: I am not clear here how to proceed if I will perform ISH immediately versus if I will store the samples for later use. Do all samples get treated with the EtOH dehydration series? Or only those that will be stored? Where do I jump into the steps listed under 3.11 if I don’t store the samples? Also, I think that “dehydrate” is clearer than “equilibrate” the pupa halves.
Response: This section is now 3.12. The dehydration step can be skipped if you will perform ISH immediately. The statement in Step 83 now reads “The fixed samples may be used to perform ISH immediately (jump to Step 109) or stored for later use”.
- Step 87: Is this the first step that requires the gel slick coating, or is this also necessary for the steps above?
Response: The gel slick coating prevents the wings from sticking to the wells of the glass-viewing dish. Gel slick coating is required during wings processing and during wings ISH.
- Step 92: Would “rehydrate” be more accurate than “equilibrate” for this step? To avoid any confusion so the reader realizes the solutions are the opposite of step 83?
Response: We have changed equilibrate to rehydrate in Step 92.
- Step 94: It might be clearer to specify to perform a second fixation of the tissues in this step, as they were already fixed once. Also, just to check, this fixation buffer is the same as the first fixation buffer?
Response: We have edited the statement in Step 94. Yes, it is the same fixation buffer as the first but different fixation buffers for wings and abdomen ISH.
- Step 97: Can the authors specify which samples/stages should only be incubated for 10 min. with Proteinase K, as opposed to wings and abdomens at P11-P15 which should be incubated for 20 min? Is it possible to incubate with Proteinase K for too long?
Response: We have successfully incubated the tissues in Proteinase K for 10 mins for P6 to P10. Prolonged incubation in Proteinase K can cause tissue degradation.
- Step 115-117: Is this meant to be done in an oven at 65C, since the samples are in a glass-viewing dish?
Response: Yes, it is fine to incubate the samples in a glass-viewing dish at 65C. Use a dedicated oven to avoid contamination.
- Step 122 & 127: Authors need to add “in a separate Eppendorf tube” to be clear the reagents are mixed-up in a different tube than what the samples are in.
Response: We added it.
- Day 3/Step 124-125: Is this at RT or at 4C?
Response: All day 3 steps are at room temperature. We have added “at room temperature” to steps 124 & 125.
- Step 126: Is pupal case separation performed in the Eppi? Or are the samples transferred to a glass view-dish, separated under a dissecting microscope, and then transferred back into an Eppi for staining?
Response: The samples are transferred back into a glass-viewing dish on day 3. We have included this sentence in step 124.
- Step 130: So is staining performed in an Eppi? Is signal development (presumably checked under a dissecting microscope) clear in an Eppi? Or should samples be in a glass view-dish?
Response: The samples are transferred back into a glass-viewing dish on day 3. We have included this sentence in step 124.
- Step 133: The imaging protocol can be found in Appendix B, was not at first apparent from the phrasing here. Also, shouldn’t this CRITICAL STEP be part of Appendix B?
Response: The CRITICAL STEP has been moved to Appendix B.
- Section 3.12 Timing: This is a useful summary, but can it be divided into two sections for Probe-preparation and Dissection/ISH with a time-estimate for that part? Also, technically for Steps 65-84 the flies have to be crossed/flipped and staged so the preparation for pupal collection takes longer than 4h.
Response: The timing section has been divided as suggested. We did not consider the waiting period for the pupal to reach the right stage because this depends on the pupal stage the user is interested in. Also, we edited step 65-84 and wrote in parenthesis “This excludes the time for larva collection and the periods of pupal development”.
- Figures 4-8: Can the authors include a sense control image for one or more of these figures? I think this actually should also be mentioned clearly in the probe synthesis and design sections as well, that controls are absolutely necessary to test staining specificity. I presume the authors use a sense control, but are there any other controls the authors regularly use and recommend for someone trying this protocol? For example, in Figure 7 can the authors show a control at P10 with a non-specific probe or without a probe to show the spot pattern is only evident in the ISH? In the legend for Figure 7 the authors should also mention that (A) and (B) are the dorsal and lateral cut view.
Response: Figure 7 legend has been adjusted. Traditionally, people did a sense control but we stopped using the sense probe since it was discovered that there can be microRNAs that can bind to the sense probe. There is no perfect control, however, we use genes that are not expressed as negative controls, and the genes that are expressed as positive controls (e.g., the yellow gene at P10 in D. guttifera). We did not take pictures of our negative controls.
- Part 6. Reagents Setup, amp agar plates: Authors should include the step of pouring the amp + agar solution into 10 cm plates and letting them harden. Also proper storage of the plates until use.
Response: We have included the agar pouring and storing steps. The statement added is “Swirl the medium to mix before pouring; be careful not to introduce bubbles. Pour into the medium-sized Petri dish, until it covers the bottom, approximately 30 mL. Store at 4 °C for 3 weeks”.
- Appendix B: This imaging protocol should be clear in case the reader has never imaged ISH samples before. In step 1, what kind of well? In a glass view-dish or a depression microscope slide? Step 2: chances are people will not have the exact imaging set-up the authors use. What is critical about the objective and the camera specs to get appropriate images? Step 5/6: Can these steps also be performed with Image J/Fiji or with other image editing software? I do not have Helicon Focus or any Adobe software in my lab, for example, and we use Image J and Affinity products.
Responses:
In step 1, what kind of well? In a glass view-dish or a depression microscope slide?
- A glass-viewing dish.
Step 2: chances are people will not have the exact imaging set-up the authors use. What is critical about the objective and the camera specs to get appropriate images?
- Any imaging stereoscope can be used.
Step 5/6: Can these steps also be performed with Image J/Fiji or with other image editing software? I do not have Helicon Focus or any Adobe software in my lab, for example, and we use Image J and Affinity products.
- Any imaging software can be used, as long aas the user conforms to the ethics of image editing.
- Appendix C: I would reference the materials list as many people are probably not familiar with gel slick reagent.
Response: Done.
- Appendix D: Step 5 – I presume the pupal case needs to be teased open before taking the pupa out by the head, and this should be added to the protocol. Do the authors also split the case a bit further? In my experience, the case needs to be about ½ open before I can pull a pupa out intact without just merely pulling off the head. Step 16 – It needs to be clear here how this protocol connects to steps 1-15 above, I presume step 16 comes after step 5, right? Also, can the authors please be more specific with “dissect pupal wings”? How do they do this? I can imagine using a surgical scissors to just cut-off the wing and a bit of the surrounding epidermis. Also for step 17, by “allow the wings to inflate” do the authors mean just incubate the dissected wings in buffer for 5-10 minutes until they have filled with buffer and inflated?
Responses:
Appendix D: Step 5 – I presume the pupal case needs to be teased open before taking the pupa out by the head,
- Yes, a little bit of the anterior dorsal part gets removed.
and this should be added to the protocol. Do the authors also split the case a bit further?
- Not necessarily.
In my experience, the case needs to be about ½ open before I can pull a pupa out intact without just merely pulling off the head.
- It may depend on the person handling the pupae.
Step 16 – It needs to be clear here how this protocol connects to steps 1-15 above, I presume step 16 comes after step 5, right?
- Yes, step 16 would come after step 5, but it should be distilled water, not PBS. Actually, these older pupae can be placed on double-sided tape, then pulled out, then submerged in water, which may work better.
Also, can the authors please be more specific with “dissect pupal wings”? How do they do this?
- Cut the wings off the thorax with surgical scissors. They swim for a minute or less in distilled water and inflate automatically, and when they reach the desired inflation level, they are fixed.
